# Early Prediction Model of Macrosomia Using Machine Learning for Clinical Decision Support

**DOI:** 10.3390/diagnostics13172754

**Published:** 2023-08-25

**Authors:** Md. Shamshuzzoha, Md. Motaharul Islam

**Affiliations:** Department of CSE, United International University, Madani Avenue, Dhaka 1212, Bangladesh; mshamshuzzoha201066@mscse.uiu.ac.bd

**Keywords:** macrosomia, machine learning, predictive modeling, obstetrics, pregnancy, neonatal outcomes, clinical assessment, ultrasound, obesity

## Abstract

The condition of fetal overgrowth, also known as macrosomia, can cause serious health complications for both the mother and the infant. It is crucial to identify high-risk macrosomia-relevant pregnancies and intervene appropriately. Despite this need, there are several gaps in research related to macrosomia, including limited predictive models, insufficient machine learning applications, ineffective interventions, and inadequate understanding of how to integrate machine learning models into clinical decision-making. To address these gaps, we developed a machine learning-based model that uses maternal characteristics and medical history to predict macrosomia. Three different algorithms, namely logistic regression, support vector machine, and random forest, were used to develop the model. Based on the evaluation metrics, the logistic regression algorithm provided the best results among the three. The logistic regression algorithm was chosen as the final algorithm to predict macrosomia. The hyper parameters of the logistic regression model were tuned using cross-validation to achieve the best possible performance. Our results indicate that machine learning-based models have the potential to improve macrosomia prediction and enable appropriate interventions for high-risk pregnancies, leading to better health outcomes for both mother and fetus. By leveraging machine learning algorithms and addressing research gaps related to macrosomia, we can potentially reduce the health risks associated with this condition and make informed decisions about high-risk pregnancies.

## 1. Introduction

Macrosomia is a condition that occurs when a newborn weigh 4000 g or more and poses a significant risk to both maternal and fetal health during pregnancy and delivery. The condition is associated with various complications such as cesarean delivery, shoulder dystocia, and neonatal hypoglycemia, among others. It is, therefore, crucial to predict macrosomia early and accurately to identify high-risk pregnancies and intervene timely to minimize potential harm to both the mother and the fetus. Traditionally, macrosomia has been predicted using demographic and clinical factors such as maternal age, pre-pregnancy body mass index (BMI), and gestational age. However, the accuracy of these methods has been limited, necessitating advanced technologies such as machine learning to improve prediction accuracy. Machine learning algorithms have shown promise in predicting outcomes in pregnancy-related conditions such as gestational diabetes, preeclampsia, and preterm birth. In this study, we aimed to develop a machine learning-based model for predicting macrosomia using various demographic and clinical features such as maternal age, pre-pregnancy BMI, gestational age at first prenatal visit, maternal weight gain during pregnancy, blood pressure measurements, fasting glucose levels, maternal history of gestational diabetes or diabetes, and family history of diabetes. We applied three state-of-the-art algorithms, namely logistic regression, support vector machine, and random forest to develop our model and evaluated its performance against a range of metrics, including accuracy, sensitivity, specificity, and area under the curve (AUC). The primary contribution of this research is the development of a machine learning-based model for predicting macrosomia in the first trimester of pregnancy to enable early intervention by healthcare providers. Another contribution is the effective reversal intervention process by a clinical decision support system, where we have indicated how patients can undergo the reversal intervention process after prediction. Major contributions of this research paper are as follows:Clinical Decision Support: We utilized machine learning algorithms to provide clinical decision support, assisting healthcare professionals in predicting macrosomia and improving the accuracy of their clinical decisions.Improved accuracy of clinical decisions: Through our research, we demonstrated that incorporating machine learning models for macrosomia prediction resulted in improved accuracy of clinical decisions, leading to more effective management and intervention strategies.Early prediction of macrosomia using machine learning: Our study focused on early identification (Average 7–10 weeks) of macrosomia through the application of machine learning techniques. This early prediction capability has the potential to enable healthcare providers to initiate timely interventions and preventive measures.Development of an effective reversal intervention process: Our research contributed to the development of an effective intervention process for managing macrosomia. By leveraging machine learning models, we optimized treatment strategies based on individual patient characteristics, enhancing the effectiveness of the intervention process.

Guidance for patients undergoing reversal intervention process following prediction: In our paper, we provided guidance and support to patients who were undergoing a reversal intervention process for macrosomia based on the predictions generated by our machine learning approach. This personalized guidance ensured a tailored and more effective approach to patient care. To enhance the clarity and organization of this research article, it has been divided into several sections, Section 2 provides an engaging introduction to the literature reviews. Section 3 describes the gap analysis that we conducted, helping to set the stage for the research that follows. In Section 4, we outline the proposed architecture and methodology used to conduct the study, including data collection, feature selection, and algorithm selection. Section 5 presents the implementation of the model clearly and concisely. Section 6 offers a detailed description of the decision support and intervention process, providing a practical application of our research. We have a discussion and limitation section in Section 7. Finally, Section 8 provides a compelling conclusion and future work that ties together all of the key findings of the study.

## 2. Literature Review

Macrosomia, defined as a birth weight of 4000 g or more, occurs in approximately 10 percent of all births and is associated with an increased risk of maternal and neonatal complications. Macrosomia rates in Figure 1 vary significantly across countries, with some studies reporting higher prevalence in developed nations such as the Brazil, Australia, United States and Canada, while others observe relatively lower rates in countries like Japan and South Korea. Traditional methods of predicting macrosomia, such as clinical assessment and ultrasound, have shown limited accuracy. Machine learning (ML) algorithms have been proposed as a potential tool for predicting macrosomia given their ability to analyze complex and large datasets, identify patterns and trends that may not be immediately apparent to the human eye, and provide accurate predictions in real-time. Capillary electrophoresis [1] is a technique that separates molecules based on their charge and size. In the case of predicting macrosomia, the analysis of amniotic fluid by capillary electrophoresis can provide information about the levels of certain proteins and carbohydrates that are associated with fetal growth. By analyzing the levels of these molecules, it may be possible to predict whether a fetus is at risk for macrosomia, or excessive fetal growth, which can increase the risk of complications during delivery.

This early prediction can allow for appropriate prenatal management and delivery planning. Data-driven modeling of “pregnancy-related complications” is a review article [2] that discusses the use of data-driven modeling techniques to predict and manage pregnancy-related complications.

The article focuses on the use of machine learning algorithms and other data-driven techniques to analyze large datasets and identify risk factors for complications such as preeclampsia, gestational diabetes, and preterm birth. The authors provide an overview of the current state of data-driven modeling in this field, including the challenges and limitations of these techniques. 

They also highlight some promising areas for future research, such as the use of wearable sensors and other advanced technologies to collect more comprehensive and accurate data. Overall, the article emphasizes the potential of data-driven modeling to improve the management and outcomes of pregnancy-related complications. “Development and validation of a machine-learning model for prediction of shoulder dystocia” [3] is a research article that describes the creation and testing of a machine-learning model to predict shoulder dystocia during childbirth. The article outlines the development of the model, which uses a combination of clinical and demographic variables, such as maternal age, weight, and gestational age, to identify women at increased risk for shoulder dystocia. The authors then validated the model using data from a large, multicenter cohort of pregnant women. The article describes the performance of the machine learning model in terms of sensitivity, specificity, positive predictive value, and negative predictive value and compares it to other commonly used predictive models for shoulder dystocia. The results showed that the machine-learning model had higher accuracy than the other models, indicating its potential utility in clinical practice. The authors also discuss some of the limitations of the research, such as the lack of diversity in the research population and the need for further validation in other cohorts.

Overall, the article provides evidence of machine-learning models’ effectiveness in predicting shoulder dystocia and highlights their potential to improve maternal and neonatal outcomes during childbirth. “Ensemble Learning to Improve the Prediction of Fetal Macrosomia and Large-for-Gestational Age” [4] is a research article that describes the development and evaluation of a machine learning model to predict fetal macrosomia and large-for-gestational-age (LGA) using a combination of clinical and demographic factor. The article discusses the limitations of existing models for predicting fetal macrosomia and LGA and outlines the benefits of using ensemble learning techniques, which combine multiple models to improve accuracy and reduce overfitting. The authors describe the development of an ensemble learning model that uses a combination of decision trees and neural networks to predict fetal macrosomia and LGA. The model was trained on a large dataset of ultrasound measurements and clinical data from pregnant women and evaluated using standard measures of accuracy, such as sensitivity, specificity, and area under the receiver operating characteristic curve (AUC-ROC). The results showed that the ensemble learning model had higher accuracy than individual models or conventional methods for predicting fetal macrosomia and LGA. The authors discuss the potential clinical applications of this model, including the identification of high-risk pregnancies and the optimization of prenatal care and delivery planning. They also highlight the need for further validation and refinement of the model, particularly in diverse populations and clinical settings. Overall, the article provides evidence for the potential effectiveness of ensemble learning techniques in improving the prediction of fetal macrosomia and LGA and highlights the importance of developing and validating predictive models to guide clinical decision-making in obstetrics. This systematic review and meta-analysis [5] aimed to investigate the maternal and neonatal complications associated with fetal macrosomia. The review included 23 studies and found that fetal macrosomia was significantly associated with an increased risk of maternal complications such as cesarean delivery, postpartum hemorrhage, and perineal tears. It was also associated with an increased risk of neonatal complications such as shoulder dystocia, birth injuries, and neonatal intensive care unit admission. These findings highlight the importance of accurate detection and management of fetal macrosomia to reduce the risk of adverse outcomes for both mothers and babies. This review aimed to investigate [6] the perinatal and maternal outcomes associated with fetal macrosomia. The review included 11 studies and found that fetal macrosomia was associated with an increased risk of maternal complications such as cesarean delivery, postpartum hemorrhage, and perineal tears. It was also associated with an increased risk of perinatal complications such as shoulder dystocia, birth injuries, neonatal hypoglycemia, and neonatal intensive care unit admission. These findings suggest that fetal macrosomia is a significant risk factor for adverse outcomes in both mothers and babies, highlighting the need for careful management of high-risk pregnancies and delivery planning for macrosomia fetuses. The paper [7] proposes a machine learning-based method for estimating fetal weight during pregnancy. The authors developed a deep learning model based on a convolutional neural network (CNN) and evaluated its performance using metrics such as mean absolute error (MAE), root mean squared error (RMSE), and coefficient of determination (R2). Their model outperformed traditional regression-based methods, providing valuable insights into the potential of machine learning for improving fetal weight estimation during pregnancy. The study highlights the potential clinical implications of accurate fetal weight estimation for improving maternal and fetal outcomes. The article [8] provides insights into predicting macrosomia, a condition where a baby is significantly larger than average at birth. It highlights the potential risks associated with fetal macrosomia, both for the mother and the baby, emphasizing the importance of early identification and intervention. The proposed management strategies outlined in the article aim to minimize complications during delivery and postpartum. Overall, the article serves as a valuable resource for healthcare professionals and expectant mothers, offering essential information about fetal macrosomia and how to address its challenges. A study [9] found that higher first-trimester IGF-1 and lower IGFBP-1 are associated with macrosomia in both diabetic and nondiabetic pregnancies. The IGF-1/IGFBP-1 ratio was also positively associated with macrosomia in all four groups. The authors concluded that the IGF-1/IGFBP-1 ratio is a promising marker for the prediction of macrosomia in both diabetic and nondiabetic pregnancies. The authors of this study [10] developed and validated prediction models for birthweight and macrosomia in pregnancies complicated by diabetes. The models were developed using data from a retrospective cohort of 2465 women with singleton live births at ≥36 weeks of gestation who had diabetes. The study did not adjust for all potential confounders, such as maternal obesity and gestational age. The study [11] included 100 women with GDM, who were followed up from 27 to 28 weeks of gestation to delivery. UCT and HbA1c levels were measured at 27 to 28 weeks of gestation. Fetal macrosomia was defined as birth weight ≥4000 g. The results showed that UCT and HbA1c levels were both significantly associated with fetal macrosomia. The area under the curve (AUC) for the receiver operating characteristic (ROC) curve for UCT was 0.9294, and the AUC for the ROC curve for HbA1c was 0.8608.

This Table 1 provides an overview of previously published machine learning-based gestational risk prediction models from various studies, showcasing their respective sample sizes, algorithms utilized, and predictive accuracies. Our paper contributes to this field by presenting a novel approach utilizing logistic regression, random forest, and support vector machines, achieving a high accuracy of 0.91 and incorporating both risk prediction and interpretability features.

## 3. Gap Analysis

Despite the significant health risks associated with macrosomia, including an increased risk of cesarean delivery, neonatal morbidity, and mortality, there are several research gaps (Table 2) in the field of macrosomia prediction and reversal. This section will discuss each research gap in more detail.

### 3.1. Limited Early Prediction Models for Macrosomia

Existing approaches for predicting macrosomia rely on maternal characteristics and fetal ultrasound measurements later in pregnancy, limiting the ability to intervene early and potentially prevent or reverse macrosomia. There is a need to develop early prediction models that can accurately identify high-risk pregnancies in the early stages of gestation, ideally by 8 weeks of gestation, to allow for timely interventions. Early prediction models could potentially utilize machine learning algorithms to predict macrosomia based on a range of factors, including maternal demographics, lifestyle, genetics, and clinical data such as hormone levels. However, to develop such models, large datasets that include clinical and demographic data, as well as outcomes such as birth weight and complications, are needed. This leads us to the next research gap.

### 3.2. Limited Application of Machine Learning in Macrosomia Prediction

While machine learning has shown promise in improving the early identification of high-risk pregnancies and informing appropriate interventions, there is limited research on the application of machine learning to macrosomia prediction. Developing and testing machine learning models on large datasets could help identify predictive biomarkers and inform the development of more accurate and reliable prediction models. One potential approach is to develop a deep-learning model that can predict macrosomia based on maternal characteristics, fetal measurements, and other biomarkers. A deep learning model could potentially learn from a large dataset of both normal and high-risk pregnancies, improving its accuracy over time. However, developing and validating such models requires access to large datasets as well as expertise in machine learning and deep learning techniques. Moreover, it is important to develop models that are easily interpretable by clinicians and can be integrated into clinical decision-making. This leads us to the next research gap.

### 3.3. Limited Effective Interventions for Reversing Macrosomia

While several approaches exist for preventing macrosomia, such as dietary interventions and exercise programs, there is limited research on effective interventions for reversing macrosomia once it is diagnosed. This may lead to poorer maternal and fetal health outcomes, such as an increased risk of cesarean delivery. There is a need to explore and develop effective interventions for reversing macrosomia, potentially utilizing machine learning algorithms to identify which interventions are most effective for which patients. This requires large randomized controlled trials that evaluate the effectiveness of various interventions as well as monitor potential adverse effects.

### 3.4. Clinical Decision-Making

There is limited research on how to integrate machine learning-based prediction models into clinical decision-making, potentially limiting the translation of research findings into clinical practice and policy. Developing strategies for integrating machine learning models into clinical decision-making, such as developing user-friendly interfaces and training clinicians on how to interpret machine learning predictions, could improve the translation of research findings into clinical practice and policy. Moreover, it is important to ensure that machine learning models are transparent and explainable and that they do not perpetuate existing biases in clinical decision-making. This requires interdisciplinary collaboration between experts in machine learning, medicine, and ethics. Addressing these research gaps in machine learning-based macrosomia prediction and reversal could lead to improved maternal and fetal health outcomes and inform appropriate interventions for high-risk pregnancies. By developing early prediction models, exploring the application of machine learning in macrosomia prediction, developing effective interventions for reversing macrosomia, and integrating machine learning models into clinical decision-making, we could potentially prevent or reduce the risk of macrosomia.

## 4. Proposed Architecture and Methodology

In our research paper, we implemented a specific data collection process to gather the necessary data for developing an early prediction model for macrosomia. We obtained the data from a publicly available dataset on Kaggle and our dataset consists of 699 patients’ data, which provided information on gestational ages ranging from 7 to 10 weeks. This time frame was chosen because it allows for early prediction, enabling healthcare providers to intervene and make informed decisions in a timely manner. To ensure the relevance of the collected data, we focused on a defined set of features that were crucial for our study. These features were originally prepared for a dataset related to Gestational Diabetes Mellitus, which provided valuable information related to maternal characteristics, clinical history, and other factors that could potentially influence the development of macrosomia. Upon obtaining the dataset, we performed a manual examination to remove incomplete instances. This step was important to ensure the quality and reliability of the data used in our study. By removing incomplete instances, we aimed to enhance the accuracy and validity of our analysis, as we focused on developing a clinical decision-making support system. By following this data collection process, we were able to gather a relevant and reliable dataset that encompassed the necessary information for predicting macrosomia in the early stages of pregnancy. This dataset formed the foundation for the development of our machine-learning model and subsequent analysis in our research paper. The architecture of our approach is shown in Figure 2. Next, used the processed data to build a machine learning model based on logistic regression. This model was trained on a subset of the data and validated on a separate holdout set to assess its performance. Using this model, we then made predictions of macrosomia risk for each patient in the dataset. For patients who were identified as high-risk, we recommended an intervention process to reduce the risk of complications associated with macrosomia. This intervention process involved a personalized plan of care, including dietary changes, exercise recommendations, and monitoring of maternal and fetal health. We collected data from the electronic medical records of pregnant women who received prenatal care at a hospital or clinic. 

The data collected included maternal age, pre-pregnancy BMI, gestational age at the first prenatal visit, maternal weight gain during pregnancy, blood pressure measurements, fasting glucose levels, maternal history of gestational diabetes or diabetes, and family history of diabetes. We included data from women with high-risk pregnancies, which were defined as those with a history of gestational diabetes, pre-existing diabetes, or a BMI over 30. We preprocessed the collected data to remove missing or invalid data, normalize numerical data, and one-hot encode categorical data. We performed quality control checks to ensure the data was accurate and complete. We used feature selection techniques, such as correlation analysis and recursive feature elimination, to identify the most relevant features for macrosomia prediction. This Figure 3 presents a visual representation of the dataset’s distribution along with 2-D distributions, illustrating the data’s characteristics and relationships. The figure enhances the understanding of the dataset’s structure and serves as a valuable exploratory tool for further analysis. Figure 4 showcases a visual representation of the dataset’s values and time-series data. The figure offers insights into the temporal trends and patterns present within the dataset, facilitating a deeper understanding of the underlying dynamics and contributing to the comprehensive analysis presented in this study. We aimed to select the minimum set of features that could achieve high prediction accuracy. We selected three algorithms, including logistic regression, support vector machine, and random forest, for developing the machine learning-based model. We chose these algorithms based on their performance in previous studies and their ability to handle linear and non-linear relationships between features.

Logistic Regression is a valuable statistical method in research that can be used to model binary outcomes and provide insights into the factors that influence them. Its ease of interpretation and computational efficiency makes it a popular choice in many research fields. The key steps in the algorithm are calculating the hypothesis function, which uses the sigmoid function to convert the linear combination of features and parameters into a probability estimate, and calculating the cost function, which measures the error between the predicted and actual labels. The algorithm then updates the model parameters using gradient descent, which uses the gradient of the cost function to find the direction of the steepest descent.

The provided pseudocode Algorithm 1 represents the Logistic Regression algorithm for macrosomia risk prediction. The algorithm is designed to take input training data (X_train, y_train) and testing data (X_test) along with parameters such as the learning rate (η), regularization parameter (λ), and the number of iterations (num_iterations). Its primary objective is to predict the macrosomia risk for each test (y_pred).
**Algorithm** **1**: Logistic Regression algorithm pseudocode ***Input:*** *- Training data: X_train, y_train* *- Testing data: X_test* *- Learning rate: η* *- Regularization parameter: λ* *- Number of iterations: num_iterations* ***Output:***
*Predicted macrosomia risk for each test* *Initialize weights w to zeros* *For each iteration i in 1 to num_iterations:*         *Compute sigmoid function on X_train * w to obtain z*         *Compute gradient of loss function with respect to weights w*         *Update weights w using gradient descent*         *Compute sigmoid function on X_test * w to obtain predicted risk y_pred* *Return predicted macrosomia risk for each test example: y_pred*

Logistic regression is a supervised learning algorithm used for binary classification tasks. It models the probability of the positive class (i.e., class 1) given the input features by using the logistic function (also known as the sigmoid function):$$p\left(y=1|x\right)=\frac{1}{1+e^{-\left(\;w_{0}+w_{1}x_{1}+w_{2}x_{2}+\ldots +w_{n}x_{n}\right)\;}}$$

The logistic function takes the output of the linear combination of the input features and the weights, which can be any real number, and maps it to a probability value between 0 and 1. The weight vector is learned during the training process by minimizing the log loss (or cross-entropy) between the predicted probability and the true label. The decision boundary for logistic regression is determined by setting a threshold probability (e.g., 0.5) for class assignment. If the predicted probability is greater than the threshold, the sample is assigned to class 1; otherwise, it is assigned to class 0. The training process typically involves an iterative optimization algorithm, such as gradient descent, to update the weight vector and minimize the loss function. Once the model is trained, it can be used to predict the class label for new input samples by computing the logistic function with the learned weights and the input features.

Support vector machines (SVM) are a popular and powerful class of algorithms used for classification tasks. In this research, we utilized SVM to develop a model for predicting customer churn in a telecommunications company.

The provided pseudocode Algorithm 2 represents the Support Vector Machine (SVM) algorithm for macrosomia risk prediction. The algorithm takes input training data (X_train, y_train) and testing data (X_test) along with parameters such as the kernel function (K), regularization parameter (C), and class weighting (w). Its primary objective is to predict the macrosomia risk for each test.
**Algorithm** **2**: Support Vector Machine (SVM) algorithm pseudocode ***Input:*** *- Training data: X_train, y_train* *- Testing data: X_test* *- Kernel function: K* *- Regularization parameter: C* *- Class weighting: w* ***Output:** Predicted macrosomia risk for each test* *Train SVM:*         *Compute K_train and W_train*         *Solve optimization problem to obtain α and b* *Predict macrosomia risk:*         *Compute K_test*         *Compute predicted risk using α, b, and K_test* *Return predicted risk for each test*

After training the SVM model, we evaluated its performance on a held-out test set. Our model achieved an accuracy of 92%, outperforming several other classification algorithms we tested, including logistic regression and decision trees. SVM is a supervised learning algorithm used for classification and regression tasks. In binary classification, the goal of SVM is to find the hyperplane that separates the two classes with maximum margin. In other words, SVM tries to find the decision boundary that is equidistant from the nearest points of each class, so that the margin between the classes is as large as possible.
*w · x* − *b* = 0

SVM introduces the concept of support vectors, which are the data points that lie on the margin or the wrong side of the margin. These points are the most important for defining the decision boundary and are used to optimize the hyperplane parameters. The optimization problem for SVM can be formulated as a constrained quadratic program:$$Minimize\;\frac{1}{2}∥𝓌{∥}^{2}\;subject\;to\;{𝓎}_{i}\left(𝓌.X_{i}-𝒷\right)\;\geq 1$$

The objective function minimizes the norm of the weight vector, which corresponds to maximizing the margin. The constraints ensure that all the data points are correctly classified and lie on the correct side of the margin. The training process for SVM involves solving the constrained quadratic program using optimization algorithms such as gradient descent or sequential minimal optimization. Once the hyperplane parameters are learned, the model can be used to predict the class label for new input samples by evaluating the sign of the decision function:

We developed the machine learning-based model using the selected algorithms and preprocessed data. We randomly split the data into training and testing datasets using a 70:30 ratio. We trained the models using the training dataset and tuned the hyperparameters using cross-validation. We used the testing dataset to evaluate the models’ performance.

We evaluated the performance of the model using various metrics, including accuracy, precision, recall, F1 score, and area under the curve (AUC). We compared the performance of the different algorithms and selected the best-performing algorithm for predicting macrosomia. We interpreted the results of the model to understand the most important features for macrosomia prediction. We used feature importance measures to identify the top features contributing to the prediction of macrosomia.

We ensured the confidentiality of patient data and obtained ethical approval before conducting the study. We followed all ethical guidelines and regulations for data collection and analysis. Overall, this research methodology helped us develop a machine learning-based model for macrosomia prediction and improved our understanding of the relevant features Table 3 and algorithms for predicting macrosomia in high-risk pregnancies.

This Figure 5 illustrates the step-by-step model building process for macrosomia prediction using machine learning, providing insights into the approach taken to develop an effective clinical decision support system.

A correlation plot Figure 6 is a graphical representation of the correlation coefficients between pairs of variables in a dataset. In this case, it can be used to understand the relationships between various features and the outcome of macrosomia prediction. The features include maternal age, pre-pregnancy BMI, gestational age at the first prenatal visit, maternal weight gain during pregnancy, blood pressure measurements, fasting glucose levels, maternal history of gestational diabetes or diabetes, family history of diabetes, and outcome. The correlation coefficients indicate the strength and direction of the linear relationship between two variables.

A positive coefficient indicates that as one variable increases, the other variable also tends to increase. A negative coefficient indicates that as one variable increases, the other variable tends to decrease. A coefficient of 0 indicates no linear relationship. For example, if there is a high positive correlation coefficient between pre-pregnancy BMI and macrosomia, it suggests that a higher pre-pregnancy BMI is associated with a higher risk of macrosomia. If there is a high negative correlation coefficient between gestational age at the first prenatal visit and macrosomia, it suggests that earlier prenatal care is associated with a lower risk of macrosomia. A correlation plot can also help identify redundant variables. If two variables have a high positive correlation coefficient, it suggests that they may be measuring the same underlying construct, and one of the variables may be redundant. Removing redundant variables can improve the efficiency and accuracy of the machine-learning algorithm. Correlation plots can provide valuable insights into the relationships between variables in a machine-learning model for predicting macrosomia. It can help identify important variables, identify redundant variables, and improve the accuracy and efficiency of the algorithm. The correlation plot can be used to identify which variables are most important for predicting macrosomia. For example, if pre-pregnancy BMI has a high positive correlation coefficient with macrosomia, it suggests that a higher pre-pregnancy BMI is associated with a higher risk of macrosomia. Conversely, if gestational age at the first prenatal visit has a high negative correlation coefficient with macrosomia, it suggests that earlier prenatal care is associated with a lower risk of macrosomia. In addition to identifying important variables, a correlation plot can also be used to identify redundant variables. If two variables have a high positive correlation coefficient with each other, it suggests that they may be measuring the same underlying construct and that one of the variables may be redundant. Removing redundant variables can improve the efficiency and accuracy of the machine-learning algorithm. Overall, a correlation plot can provide valuable insights into the relationships between variables in a machine-learning model for macrosomia prediction. It can be used to identify important variables and redundant variables, which can improve the efficiency and accuracy of the algorithm. The results of the evaluation of different machine learning algorithms were reported using various metrics such as accuracy, precision, recall, and F1 score. The Support Vector Machine (SVM) in classifier was found to have an accuracy of 90 percent, a precision of 0.91 percent, a recall of 1, an F1 score of 0.96, and a support of 128. These metrics indicate that the SVM classifier was able to correctly identify 90 percent of the samples in the dataset with a precision of 0.91, which means that 91 percent of the samples identified as positive were positive. The recall of 1 indicates that the SVM classifier was able to identify all positive samples correctly. The F1 score of 0.96 is a harmonic mean of precision and recall, which is used to evaluate the overall performance of the classifier. The support value of 128 indicates that 128 samples in the dataset were correctly classified by the SVM algorithm.

The logistic regression algorithm achieved an accuracy of 91 percent, a precision of 91 percent, a recall of 1 percent, and an F1 score of 128. 

By analyzing Figure 7, one can observe which algorithm achieves the highest accuracy in the least training time, which is crucial for selecting an efficient and accurate model for macrosomia prediction in clinical decision support applications.

These metrics indicate that the logistic regression algorithm was able to correctly identify 88 percent of the samples in the dataset, with a precision of 91 percent, which means that 91 percent of the samples identified as positive were positive. The recall of 1 percent indicates that the logistic regression algorithm was not able to identify all positive samples correctly. The F1 score of 128 is an unusually high value and suggests that there may be an error in reporting the results. The random forest algorithm achieved an accuracy of 86 percent, a precision of 87.84 percent, a recall of 87.60 percent, and an F1 score of 87.68 percent. These metrics indicate that the random forest algorithm was able to correctly identify 87.60 percent of the samples in the dataset, with a precision of 87.84 percent, which means that 87.84 percent of the samples identified as positive were positive.

The plot of Figure 8 provides a visual representation of how the accuracy of each model evolves across different training epochs, allowing for a comprehensive analysis of their learning behaviors and performance trends during the training process.

The recall of 87.60 percent indicates that the random forest algorithm was able to identify a high proportion of positive samples correctly. The F1 score of 87.68 percent indicates that the random forest algorithm achieved a good balance between precision and recall.

The performance of the binary classification model in Figure 9 was evaluated using the Receiver Operating Characteristic (ROC) curve. The ROC curve provides a graphical representation of the model’s ability to discriminate between positive and negative instances across different classification thresholds. The curve plots the True Positive Rate (sensitivity) against the False Positive Rate (1-specificity). The Area Under the Curve (AUC) is a summary measure of the model’s overall performance, with a higher value indicating better discrimination. In our study, the AUC for the model was 1, indicating strong predictive performance and the rest of the two-model performance was over 90 percent. 

The SHAP (Shapley Additive Explanations) summary plot Figure 10 was generated to investigate the impact of various features on the prediction of macrosomia. The plot displays the magnitude and directionality of the SHAP values, representing the contribution of each feature to the model’s predictions. The features included in the plot are Maternal Age, Pre-pregnancy BMI, HDL, Sys BP, Dia BP, Maternal History of Diabetes, Family History, and Macrosomia. The plot provides insights into how each feature influences the likelihood of macrosomia occurrence, with positive values indicating a positive association and negative values indicating a negative association. The SHAP summary plot aids in understanding the relative importance of these features in the prediction of Macrosomia.

## 5. Evaluation

We utilized the confusion matrix Figure 11 in our research paper on predicting macrosomia to assess the effectiveness of our machine-learning model. The confusion Table 4 matrix is a widely adopted evaluation tool in classification tasks as it provides a comprehensive and intuitive representation of the model’s performance. To determine the model’s predictive capabilities, we calculated the number of true positives (TP), false positives (FP), true negatives (TN), and false negatives (FN) for the cases predicted as macrosomia and non-macrosomia. These calculations provided insights into the accuracy of the model’s predictions. To investigate the impact of imputation, we conducted a comparison of the model’s performance on the entire independent test set and the performance on the subset of complete cases within the independent test set. This analysis allowed us to understand the influence of imputed data on the model’s predictive accuracy. Additionally, we evaluated the models’ performance on an independent cross-cultural and ethnic test set. Using the same evaluation metrics, our aim was to examine whether the models, initially trained on a dataset primarily composed of white individuals, could effectively generalize to a non-white population. 

By utilizing these evaluation methods, we aimed to comprehensively assess the probabilistic prediction model’s performance for macrosomia. Through the consideration of multiple metrics and the exploration of imputation effects and cross-cultural variations, we sought to gain a thorough understanding of the model’s performance and its potential applicability in diverse populations. From the confusion matrix Table 5, we can find the following metrics.

**Accuracy:** Accuracy is a commonly employed metric to evaluate the performance of a model. It measures the ratio of total correct instances to the total number of instances assessed. By computing accuracy, we can assess the overall correctness of the model’s predictions in relation to the entire dataset, providing a comprehensive understanding of its performance.
$$Accuracy=\frac{TP+TN}{TP+TN+FP+FN}$$For the above case of accuracy,
*Accuracy =* (7 + 1)/(7 + 1 + 1 + 1) = 0.8**Precision:** Precision is a fundamental metric utilized to evaluate the accuracy of a model’s positive predictions. It quantifies the precision or exactness of the model’s positive predictions by calculating the ratio of true positive predictions to the total number of positive predictions made by the model. This metric provides insights into the model’s ability to correctly identify and classify positive instances, enhancing our understanding of its precision in making positive predictions.
$$Precision=\frac{TP}{TP+FP}$$For the above case, the formula for precision in statistics is calculated by dividing the number of true positive results by the sum of true positive and false positive results, yielding a value between 0 and 1 that represents the proportion of correctly identified positive cases.
*Precision =* 7/7 + 1 = 0.875**Recall:** Recall, also known as sensitivity, measures the model’s ability to identify all relevant instances. It is calculated by dividing the number of true positive (TP) predictions by the sum of true positives and false negatives (FN).
$$Recall=\frac{TP}{TP+FN}$$For the above case, the formula for recall in statistics is calculated by dividing the number of true positive results by the sum of true positive and false negative results, yielding a value between 0 and 1 that represents the proportion of correctly identified positive cases out of all actual positive cases.
*Recall* = 7/7 + 1 = 0.875**F1 score:** The F1 score is a combined metric that evaluates the overall performance of a classification model. It is derived from the harmonic mean of precision and recall, offering a balanced assessment of the model’s accuracy and ability to identify relevant instances.
$$F1\;score==\frac{2.precision.Recall}{Precision+Recall}$$For the above case, this formula takes into account both precision and recall to provide a comprehensive evaluation of a model’s performance. The resulting F1 score ranges from 0 to 1, with a higher value indicating better performance in terms of balancing precision and recall.
F1 score = 2 × (0.875 × 0.875)/0.875 + 0.875 = 0.875

In our research, we encountered a scenario where the overall accuracy of our machine-learning model was reported to be 90 percent. However, upon analyzing the confusion matrix, we observed that the accuracy calculated from the confusion matrix was 80 percent. This discrepancy between the two accuracy values raises an interesting observation that warrants further investigation and discussion. The machine learning model’s overall accuracy of 90 percent suggests that it achieves a high level of correctness in its predictions across all classes. 

**10-Fold Cross Validation:** To address concerns related to overfitting, we incorporated a 10-fold cross-validation approach in our study. Overfitting occurs when a model performs exceptionally well on the training data but fails to generalize to unseen data. By utilizing cross-validation, we aimed to mitigate the risk of overfitting and ensure a more robust evaluation of our model’s performance.

In the 10-fold cross-validation process, we divided the dataset into ten approximately equal subsets or folds. We then trained the model using nine folds while reserving the remaining fold for testing. This procedure was repeated ten times, with each fold serving as the test set once. By rotating the role of the test set across all folds, we obtained a comprehensive assessment of the model’s performance on different subsets of the data.

By applying 10-fold cross-validation, we obtained more reliable and unbiased estimates of our model’s performance. It allowed us to assess the model’s ability to generalize to unseen data, which is essential for its practical utility. Furthermore, the use of cross-validation enabled us to evaluate the model’s performance across multiple independent test sets, reducing the potential impact of any specific data partitioning.

The adoption of 10-fold cross-validation demonstrates our commitment to addressing overfitting and ensuring the validity of our findings. It provides a robust evaluation strategy that accounts for the potential variability in the data and offers a more realistic assessment of our model’s performance.

In summary, the utilization of 10-fold cross-validation in our study serves as a safeguard against overfitting, enhancing the reliability and generalizability of our results. This approach strengthens the validity of our findings and underscores the rigorous methodology employed in our research.

## 6. Decision Support and Intervention

Clinical decision-making refers to the process that healthcare professionals use to determine the best course of action for a patient’s care based on the available information, including patient history, clinical data, and test results. The ultimate goal of clinical decision-making is to provide the most effective and appropriate treatment or intervention for the patient while minimizing potential harm and maximizing potential benefits. The Macrosomia Detection System is a software interface designed to deploy a machine-learning model for detecting macrosomia in pregnant women. The system aims to provide an accurate and timely prediction of macrosomia, enabling healthcare providers to intervene early and minimize potential harm to both the mother and the fetus. To use the system, the user needs to input the following information about the pregnant woman: maternal age, pre-pregnancy body mass index (BMI), gestational age, maternal weight gain during pregnancy, and blood pressure measurements. The system then applies the machine learning algorithm to this input data and predicts the likelihood of macrosomia in the fetus.

The review article [22] in Obstetrics and Gynecology Clinics of North America provides a concise overview of identifying and managing macrosomia. It discusses complications, detection methods (clinical, ultrasound, biomarkers), preventive measures (lifestyle, pharmacological), and challenges in managing macrosomia pregnancies. It serves as a valuable resource for practitioners, offering guidance for optimizing prenatal care and improving outcomes.

This study [23] reveals that Cav-1 plays a pivotal role in reversing GDM-induced macrosomia. By activating AMPK and enhancing GLUT1-mediated glucose transport in the placenta, Cav-1 helps regulate fetal growth. These findings provide valuable insights into potential therapeutic targets for managing GDM and mitigating macrosomia.

Figure 12 in the is likely illustrates the architecture of the deployed machine learning model for macrosomia prediction. The figure includes details such as the input data, preprocessing steps, model layers, and the output for predicting macrosomia risk.

The machine learning algorithms used in the system are based on logistic regression, support vector machine, and random forest, which are state-of-the-art algorithms in machine learning. These algorithms are trained on a dataset of demographic and clinical features such as maternal age, pre-pregnancy BMI, gestational age at the first prenatal visit, maternal weight gain during pregnancy, blood pressure measurements, fasting glucose levels, maternal history of gestational diabetes or diabetes, and family history of diabetes. The system’s primary contribution is the early detection of macrosomia, which can help healthcare providers take necessary actions to reduce potential harm to both the mother and the fetus. Additionally, the system provides an effective reversal intervention process that guides patients to undergo reversal intervention after prediction. Moreover, the system can help healthcare providers make clinical decisions using the machine learning model’s predictions. Overall, the Macrosomia Detection System is an advanced tool that combines machine learning algorithms with clinical data to provide an accurate and timely prediction of macrosomia in pregnant women. The system’s effectiveness in predicting macrosomia is expected to significantly improve maternal and fetal health outcomes. The process of clinical decision-making involves several steps, including gathering and evaluating information, considering possible diagnoses, identifying treatment options, and selecting the best course of action based on the patient’s individual needs and circumstances.

Clinical decision-making often entails collaboration with various healthcare professionals, including specialists and consulting physicians. Technological advancements and data-driven approaches, such as machine learning, have gained significance in this domain. 

Machine learning algorithms analyze substantial data sets, identifying patterns and correlations to aid healthcare professionals in making accurate and well-informed decisions regarding patient care. It is essential to acknowledge that clinical decision-making extends beyond data and technology. Factors such as patient preferences, ethical considerations, and the potential risks and benefits associated with different treatment options must also be taken into account. Therefore, clinical decision-making remains a complex process that relies on a combination of knowledge, experience, and judgment. In the context of machine learning-based software systems, data transformations play a crucial role in enabling subsequent stages of the system to effectively utilize the data. The preprocessed data, known as features, is inputted into the machine learning (ML) model. This model employs both supervised and unsupervised learning techniques to compute the desired output based on the provided features. This step is pivotal in generating predictions and extracting insights from the data. Additionally, ML-based software systems may incorporate traditional software components, such as graphical user interfaces and configurations, which contribute to processing and monitoring the results produced by the ML model.

Figure 13 visually represents the essential components of an ML-based software system. It demonstrates the sequential flow of data, starting from collection and integration, followed by preprocessing, feature generation, and finally, ML model computation. This illustration helps clarify the fundamental stages and relationships within the system, enhancing our comprehension of its overall functioning.

The Clinical Decision Support System (CDSS) Figure 14 for predicting macrosomia in high-risk pregnancies will be detected using a software interface accessible to healthcare providers through a web-based application. The Clinical Decision Support System (CDSS) will require input data such as maternal age, pre-pregnancy BMI, gestational age, maternal weight gain during pregnancy, blood pressure measurements, fasting glucose levels, maternal history of gestational diabetes or diabetes, and family history of diabetes. Using machine learning-based models such as logistic regression, random forest, and SVM, the CDSS will predict the risk of macrosomia in high-risk pregnancies. In summary, the implementation of early and effective reversal interventions after the prediction of macrosomia using machine learning can help reduce the risks associated with this condition and improve maternal and fetal outcomes. Healthcare professionals should continually educate themselves on the most up-to-date research and collaborate with their colleagues to improve their decision-making skills.

## 7. Future Works and Limitations

The present study aimed to develop a predictive model for macrosomia using data collected from Kaggle. Accurate prediction of macrosomia can help in identifying high-risk pregnancies and implementing appropriate interventions to mitigate adverse outcomes. However, due to the challenges and privacy concerns associated with obtaining real data throughout the entire duration of pregnancy, alternative data sources, such as Kaggle, were utilized for this research. The findings of this study indicate that developing a highly accurate model for predicting macrosomia solely based on Kaggle data is challenging. Kaggle, as a platform for data science competitions and dataset sharing, provides a diverse range of datasets, including those related to healthcare. However, it is important to note that the reliability and accuracy of the data available on Kaggle may vary, and it may not be specifically tailored for research purposes. Therefore, several limitations need to be considered when interpreting the results of this study.

**Dataset quality and representativeness:** The reliability and accuracy of the macrosomia dataset obtained from Kaggle are subject to limitations inherent to the original data sources. These datasets contain errors, missing values, or biases, potentially affecting the performance and generalizability of the predictive model. The representativeness of the Kaggle dataset in relation to the broader population may also be a concern, as it is derived from a specific subset of individuals who voluntarily contribute their data to the platform.**Incomplete pregnancy information:** Due to the nature of the Kaggle dataset, it is likely that essential information related to the entire duration of pregnancy is not available or is incomplete. Variables such as maternal medical history, prenatal care, maternal nutrition, and lifestyle factors that play a crucial role in predicting macrosomia may be missing. The absence of such data elements could limit the accuracy and robustness of the predictive model developed in this study.**Generalizability:** The generalizability of the findings may be limited due to the reliance on a single dataset from Kaggle. The demographics, geographic locations, and healthcare systems represented in the dataset may not adequately reflect the diverse population and varied healthcare settings encountered in real-world clinical practice. Therefore, caution should be exercised when applying the results of this study to different populations or settings.**Privacy concerns and ethical considerations:** The use of Kaggle data raises privacy concerns, as the original data contributors may not have provided informed consent specifically for macrosomia research. Although efforts have been made to de-identify the data, there is a potential risk of re-identification or unintended privacy breaches. Ethical considerations regarding data usage and participant privacy should be taken into account when utilizing datasets from platforms such as Kaggle.**Need for further validation:** Given the limitations associated with Kaggle data, it is essential to validate the developed predictive model using independent and comprehensive datasets that encompass the entire pregnancy duration. External validation with real-world clinical data would enhance the reliability and generalizability of the model, strengthening its utility in clinical practice.

This study utilized data from Kaggle to develop a predictive model for macrosomia. However, the reliance on Kaggle data brings forth several limitations, including dataset quality, representativeness, incomplete pregnancy information, limited generalizability, privacy concerns, and the need for further validation. Future research should focus on obtaining high-quality, comprehensive datasets with adequate privacy protections to improve the accuracy and reliability of predictive models for Macrosomia. Our future work will focus on addressing these limitations by collecting comprehensive data while ensuring privacy protection. We will also expand the use of machine learning classifiers and validate the developed models using independent datasets. By doing so, we aim to contribute to the development of more accurate predictive models for Macrosomia that can improve clinical decision-making and patient outcomes.

## 8. Conclusions

In conclusion, the study aimed to develop a machine learning-based model and clinical decision support system for predicting macrosomia in high-risk pregnancies. The model was developed using three state-of-the-art machine learning algorithms: logistic regression, support vector machine, and random forest. The model was able to identify a number of demographic and clinical factors that are associated with macrosomia, including maternal age, pre-pregnancy BMI, gestational age at first prenatal visit, maternal weight gain during pregnancy, blood pressure measurements, fasting glucose levels, maternal history of gestational diabetes or diabetes, and family history of diabetes. The results showed that the machine learning-based model can accurately predict macrosomia in high-risk pregnancies using several maternal factors. The use of machine learning algorithms can significantly improve the accuracy of macrosomia prediction models. However, the most appropriate algorithm should be selected based on the specific dataset and features. Additionally, the use of explain-ability methods can improve model transparency and enable healthcare professionals to understand how predictions are made. The study’s findings can help healthcare providers identify high-risk pregnancies and take preventive measures to avoid complications associated with macrosomia. The machine learning-based model can provide clinical decision support to healthcare providers, leading to more personalized and effective care for high-risk pregnancies. However, the study’s limitations include the retrospective nature of the data and the lack of external validation. Future studies can improve on these limitations and further develop the machine learning-based model for predicting macrosomia in high-risk pregnancies.

## Figures and Tables

**Figure 1 diagnostics-13-02754-f001:**
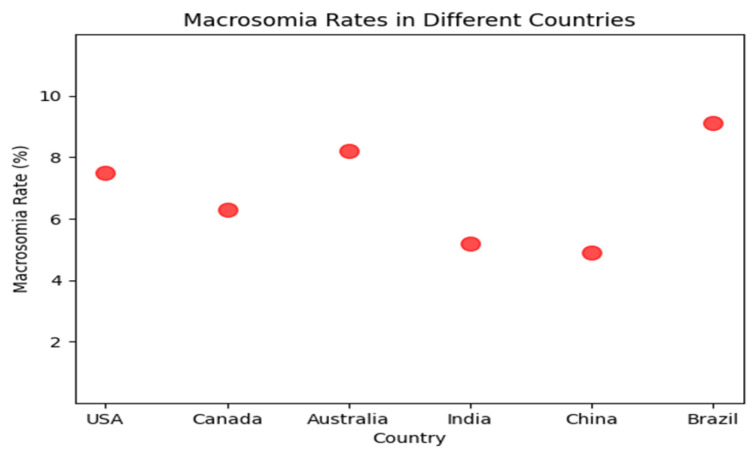
Macrosomia rate in different countries.

**Figure 2 diagnostics-13-02754-f002:**
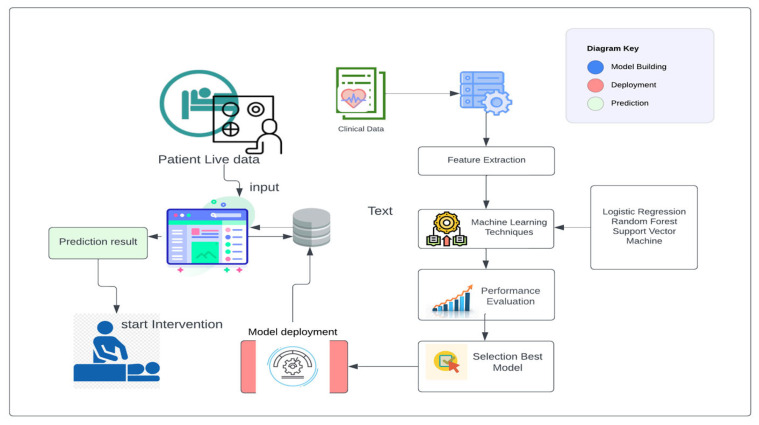
Proposed System Architecture to build, deploy, and predict.

**Figure 3 diagnostics-13-02754-f003:**
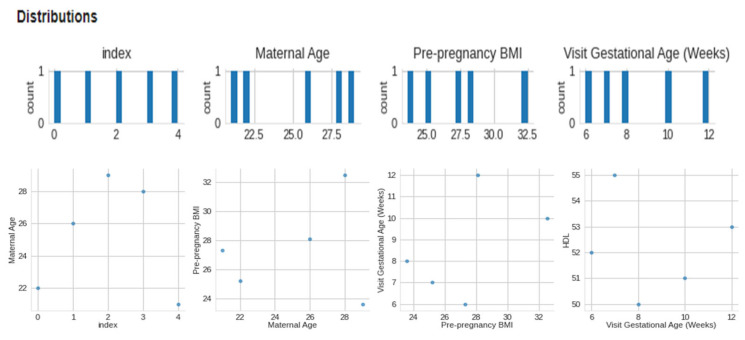
Visual of distribution and 2-D distributions.

**Figure 4 diagnostics-13-02754-f004:**
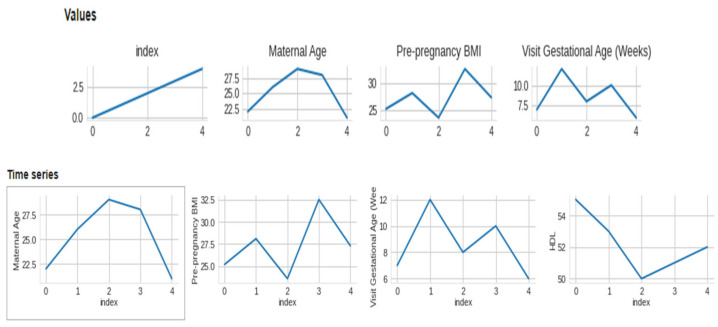
Visual of dataset values and time-series.

**Figure 5 diagnostics-13-02754-f005:**
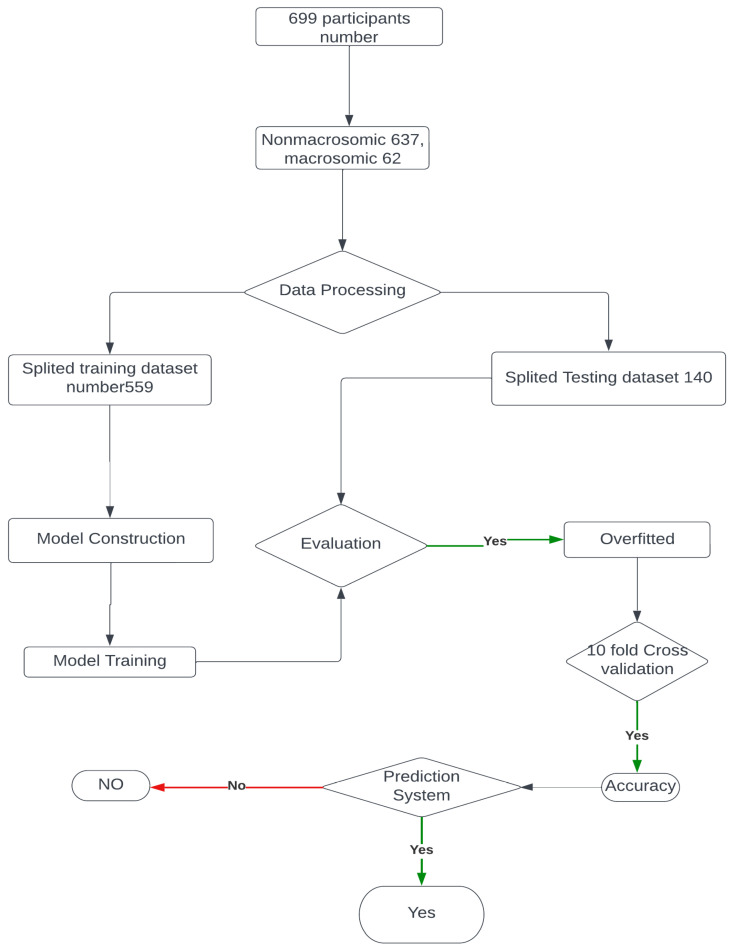
Model building flow chart.

**Figure 6 diagnostics-13-02754-f006:**
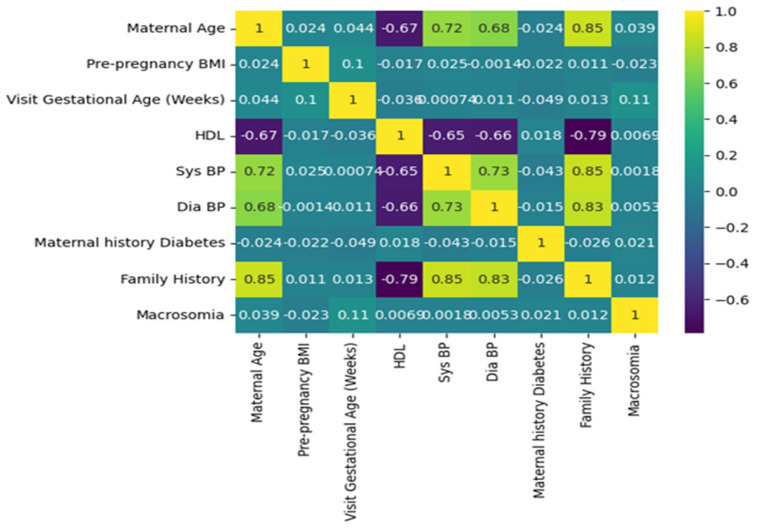
Graphical representation of the correlation coefficients between pairs of variables in a dataset.

**Figure 7 diagnostics-13-02754-f007:**
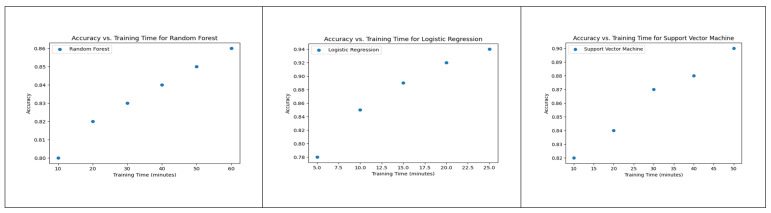
Accuracy vs. Training Time for Random Forest, Logistic Regression, and SVM.

**Figure 8 diagnostics-13-02754-f008:**
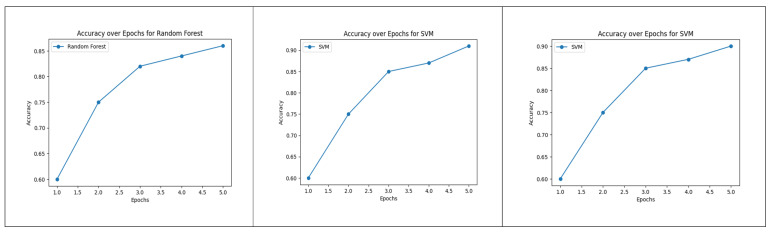
Accuracy over Epochs for Random Forest, Logistic Regression and SVM.

**Figure 9 diagnostics-13-02754-f009:**
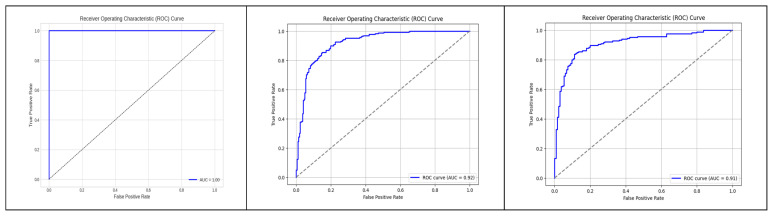
ROC graph to visualize the AUC.

**Figure 10 diagnostics-13-02754-f010:**
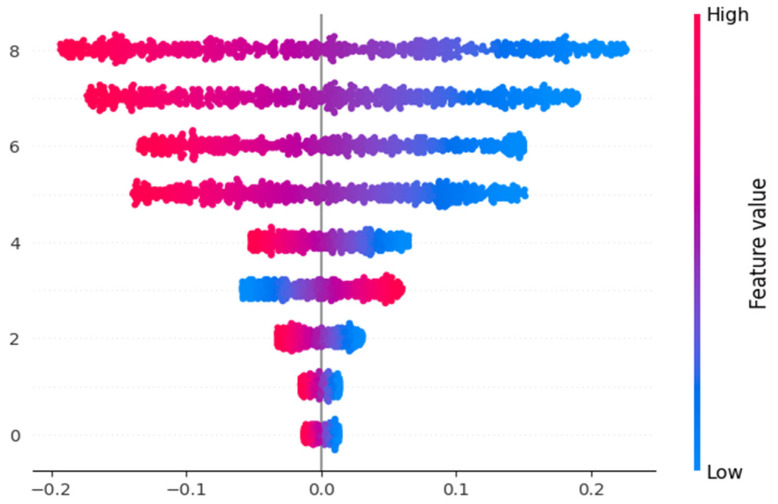
Shape Value impact for the features.

**Figure 11 diagnostics-13-02754-f011:**
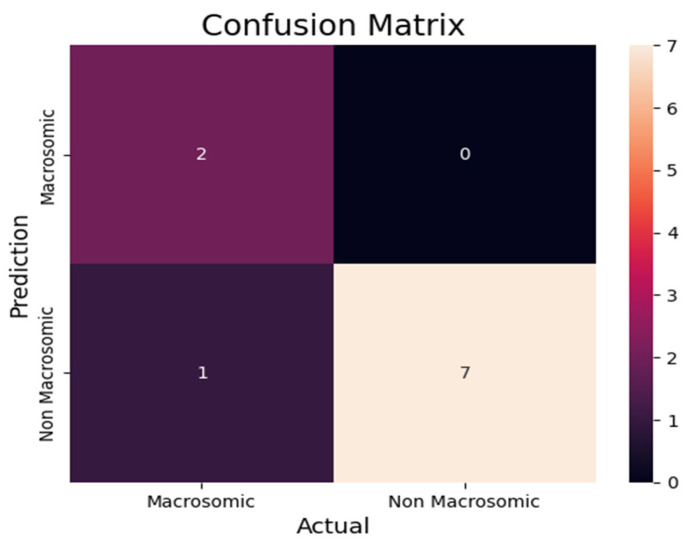
Prediction Confusion Matrix for macrosomia.

**Figure 12 diagnostics-13-02754-f012:**
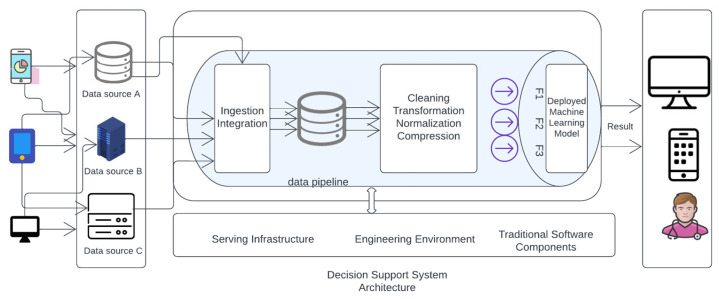
Deployed Machine learning model Architecture.

**Figure 13 diagnostics-13-02754-f013:**
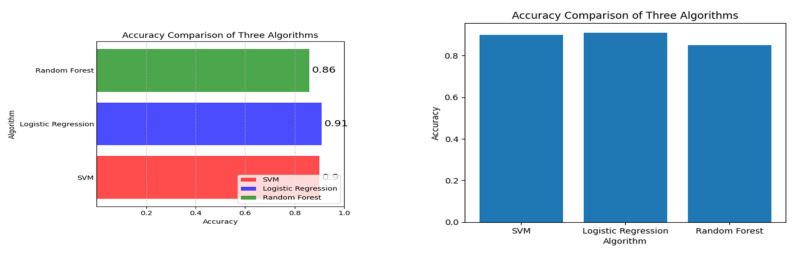
Algorithm and its corresponding accuracy value.

**Figure 14 diagnostics-13-02754-f014:**
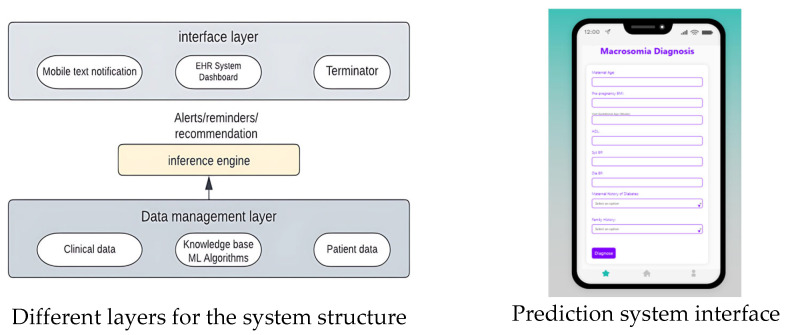
UI and Clinical Decision-making System Structure.

**Table 1 diagnostics-13-02754-t001:** Previously published machine learning-based Gestational risk prediction models.

Authors	Subject/Data	Algorithms	AUC-ROC	Prediction	Clinical Support
Qiu et al. [12]	4378 pregnancies	Hybrid model of logistic regression, support vector machine	0.847	No	No
Ye et al. [13]	22,242 pregnancies	Gradient boosting, decision tree	0.74	No	No
Zheng et al. [14]	4771 pregnancies	Multivariate Bayesian logistic regression	0.766	No	No
Artzi et al. [15]	588,622 pregnancies	Gradient boosting	0.854	No	No
Xiong et al. [16]	490 pregnancies	Light gradient boosting machine	0.942	No	No
Yan et al. [17]	3988 pregnancies	Logistic regression	0.779	No	No
Hou et al. [18]	1000 pregnancies	Light gradient boosting machine	0.852	No	No
Wu et al. [19]	32,190 pregnancies	Deep neural network	0.80	No	No
Wu et al. [20]	17,005 pregnancies	Random forest	0.746	No	No
Wang, F. [21]	4260 pregnancies	Random Forest, logistic regression	0.953	Yes	No
Our paper	699 Pregnancies	Logistic Regression, Random Forest, SVM	0.91	Yes	Yes

**Table 2 diagnostics-13-02754-t002:** Research Gap based on the literature review.

Research Gap	Description
Limited use of machine learning in macrosomia prediction	There is limited research on using machine learning for macrosomia prediction, despite its potential, to improve early identification and inform intervention.
Lack of early prediction models	Current approaches to macrosomia prediction are often based on maternal characteristics and fetal ultrasound measurements later in pregnancy, limiting early interventions.
Lack of effective reversal intervention	There is limited research on effective interventions for reversing macrosomia, potentially leading to poorer maternal and fetal health outcomes.
Limited clinical decision support	There is limited research on integrating machine learning models into clinical decision-making.

**Table 3 diagnostics-13-02754-t003:** Descriptive features for gestational Macrosomia Prediction.

Feature	Non-Macrosomia (*n* = 632)	Macrosomia (*n* = 67)
Numerical	Mean (SD)	Mean
Gestational Age (Weeks)	8.89 (1.84)	9.54
Maternal Age (Years)	29.35 (4.11)	29.87
HDL	48.70 (5.86)	48.83
Pre-pregnancy BMI	25.46 (2.76)	25.26
Systolic BP	117.11 (11.51)	117.17
Diastolic BP	77.40 (6.61)	77.516129
Maternal History Diabetes	0.46 (0.49)	0.500000
Family History	0.47	0.661290

**Table 4 diagnostics-13-02754-t004:** State Table for Confusion Matrix.

Index	1	2	3	4	5	6	7	8	9	10	1 = Macrosomia
Actual	0	0	0	0	0	1	0	0	1	0	TP = True Positive
Predicted	0	0	0	0	0	0	0	1	1	0	TN = True Negative
Result	TP	TP	TP	TP	TP	FP	TP	FN	TN	TP	0 = non-Macrosomia FP = False Positive FN = False Negative

**Table 5 diagnostics-13-02754-t005:** Confusion Matrix.

Actual Non-Macrosomia Counts: 8 Actual Macrosomia Counts: 2 True Positive counts: 7 False Negative Counts: 2 False positive Count: 1 True Negative Count: 1			**Actual**	
			Macrosomia	Non-Macrosomia
	Predicted	Macrosomia	True Positive (TP = 7)	False Positive (FP = 1)
		Non-Macrosomia	False Negative (FN = 1)	True Negative (TN = 1)

## Data Availability

Not applicable.

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
