# Peer review of "Early Prediction Model of Macrosomia Using Machine Learning for Clinical Decision Support"

_diagnostics, 2023, doi:10.3390/diagnostics13172754_

Round 1

Reviewer 1 Report

The manuscript presented Early Prediction ML Model of Macrosomia. The manuscript is well written however there are some suggestions for the author that can further improve the quality of manuscript before it is published.

1. In section 3, author should provide the detail of dataset used in this study under the heading Dataset Description.

2. Mathematical equation on line 319 is not according to the format of text that need to be corrected.

3. On line 328, author describe that dataset is partitioned into 70:30 ratio for training and testing of the Model and also used cross-validation which is confusing. Because in holdout validation, the dataset is partitioned while in cross-validation dataset is divided in k group and each group is used for training and testing purpose of the model. 

4. Author also employed AUC for validation of the model through ROC in Table 1. I would recommend that ROC graph should be included in the manuscript to visualize the AUC. 

5. How the proposed model avoids the problem of model overfitting problem. This should be described in the manuscript.  

Language of the manuscript is appropriate and need minor grammatical corrections. 

Author Response

Please find the review response in the attachment.

Reviewer 2 Report

I am really grateful to review this manuscript. In my opinion, this manuscript can be published once some revision is done successfully. I made one suggestion and I would like to ask your kind understanding. This study used numeric data from 699 patients, applied three machine learning models and achieved the accuracy score of 91% with logistic regression for the prediction of macrosomia. I would argue that this is a good start. However, it can be noted that the Shapley Additive Explanations (SHAP) summary plot is very effective to identify the direction of association between macrosomia and its major predictor. In this context, I would like to ask the authors to derive the SHAP summary plot. 

Minor editing of English language required. 

Author Response

Please find the review response in the attached file.

Reviewer 3 Report

The authors applied different ML algorithms to predict the outcome of Macrosomia. The logistic regression algorithm outperformed the other methods based on the performance measurements. The model used Kaggle databases, and justify that this is the first research model to be published based on the data. The idea is great. However, I have major concerns/suggestions:

- I could not spot the actual number of samples, I think 10 is used as an example in table 4, if yes, I believe 10 is a really small number of samples to build a prediction model.

- The explanation of how to calculate the performance measurements based on 10 samples is too much of an explanation. Usually in ML papers they just mention the equations without the need of tutoring on how to use the equations.

 - What is the split of the data, is it 70 training / 30 testing, or 10-fold cross validation?

- How the author optimized the hyper-parameters of the model, or just run the classifier on default settings.

Editing and resectioning are required.

Author Response

(The authors gave the same response as above.)

Round 2

Reviewer 3 Report

required - I think AUCROC must be drawn by running the model at different time with different parameters ( different points), I think one point is not enough.

I also suggest to add more literature. 19 references are low for such interesting topics with so many contributions.

 ensure that the data =>  ensure the data

handle both linear and non-linear => handle linear and non-linear

The authors may go through the paper again, please.

Author Response

(The authors gave the same response as above.)
